# A Comparative Analysis on Residents' Reservation Willingness for Bus Service Based on Option Price

**Xun Zheng [1],\* and Tomio Miwa [2]**

[1]  Department of Civil and Environmental Engineering, Nagoya University, Nagoya 464-8603, Japan
[2]  Institute of Materials and Systems for Sustainability, Nagoya University, Nagoya 464-8603, Japan; miwa@nagoya-u.jp
\*  Correspondence: zheng.xun@f.mbox.nagoya-u.ac.jp; Tel.: +81-52-789-5018

**Abstract:** Population decline is a pressing issue facing Japan and other developed countries. Local governments in Japan are seeking solutions to insure they meet the daily travel demands of the elderly. Although subsidy for local bus companies is a highly practical policy, a careful investigation to determine its reasonable level is required. This paper investigates the option price of local bus services, that is, the willingness of residents to pay to maintain the services and attempts to gain insights on a reasonable level of subsidy for local bus companies. A comparative analysis among age groups and different city size groups was made. The result showed that elderly residents show a higher option price value than younger and middle-age residents.

**Keywords:** aging society; local bus service; option price; contingent valuation method

## 1. Introduction

Population decline and the aging of society are two pressing, contemporary issues in Japan [1,2]. The latest Japanese annual nationwide census indicates that the total population decreased to 227,000 (−0.18%) compared to the previous year [3]. The total non-elderly population (under 65 years old) decreased by 788,000 a year, and the total elderly population (over 65 years old) increased by 1.135 million a year [3]. The birth rate has already fallen to its lowest degree, 1.26 [4], and the prospects given this trend are expected to be severe. The elderly's transportation accessibility has been identified as a key concept of sustainable society [5], and society must develop a reliable long-term system for the elderly to live and work [6–10]. Studies on aging society have discussed that transport accessibility should be a topic of concern for a sustainable society [11–13]. Consequently, for a sustainable society, the elderly's daily travel demands should be met as part of basic insurance [14]. For this reason, the evaluation of the public transport value will be a useful reference for government policy making.

Local bus services are part of the basic transportation system in Japan, and they are used for daily travel. Thus, it is necessary to keep buses in operation to support the elderly. However, local bus companies have been facing operational difficulties [15]. There are 2171 bus companies in Japan. To maintain services for residents, bus companies have been losing 22.5 billion JPY a year. Around 70% of the companies are in deficit [16]. According to Japanese policy for public transport, government subsidy from both national and local governments does not cover the total deficit [17]. This makes it difficult for local bus companies to maintain their service, which is exacerbated by an aging society.

Against this background, it is important to investigate the value of such local bus services, which could help us to determine whether the subsidy amount for local bus companies is appropriate or not. In this paper, the authors investigate the value of the Japanese local bus services based on the concept of option price (OP), which is, in turn, based on the residents' willingness to pay tax to maintain the local bus service. The stated preference (SP) survey for a hypothetical tax was conducted and a utility

function for the tax payment was estimated based on the utility maximization theory using SP data. The OP value of the local bus service was estimated from the utility function and compared among age groups and different-sized city groups, and such a comparison of OP value is the first trial to the best of the authors' knowledge.

The remaining paper is structured as follows. Section 2 reviews the relevant literature and Section 3 describes the methodology applied in the analysis. Section 4 outlines the questionnaire data and presents the summary statistics. Section 5 provides the model estimation results and subsequent OP calculation and discusses the findings. Finally, Section 6 concludes the paper and suggests future research directions.

## 2. Literature Review

Weisbroad (1968) first argued the economic value for rarely used services, such as national parks [18]. The rationale supporting this claim is that if there is even a small possibility of using such services, a willingness to pay for maintaining them would be expected. This economic value is called the option value (OV), and it is defined as the difference between willingness to pay for maintaining a target service and expected consumer surplus (CS). Here, the value of willingness to pay is called the option price (OP).

Many studies on OV and OP exist in environmental economics [19–22]. Brookshire et al. (1983) explored the analytical structure of OP and proposed a modification of the contingent valuation method to estimate it for non-market goods [19]. Smith (1985) discussed the interpretation of OP and risk in a dwelling environment [20]. Freeman III (1986) developed a utility-based model to analyze OP and OV by considering uncertainties in service use and its supply [21]. Willis (1989) discussed OV in relation to CS by taking the benefit of wildlife conservation as an example [22]. Kridel et al. (1993) analyzed the demand of telecommunication services to show an empirical example in which OV significantly improves demand estimation [23]. Boyer and Polasky (2004) reviewed literature on nonmarket valuation and classified the economic value of urban wetland, in which OV is defined as a value for potential future use [24].

In studies on transportation, research on OV and/or OP also exists. Weisbrod (1968) argued the theoretical basis of OV by taking public transportation service as an example [18]. Bristow et al. (1992) conducted a contingent valuation survey of two bus services in the U.K. to investigate various non-use values [25]. Roson (2000, 2001) discussed the appropriateness of subsidies from local government to the Italian public transportation service from the aspect of OV [26,27]. Humphrey and Fowkes (2006) conducted an SP survey on different railway service levels in Scotland and estimated the economic values of the service, including OV [28]. In their study, the utility function of railway service, which was estimated based on a discrete choice model, was used for estimating values of CS, OP, and OV. Geurs et al. (2006) estimated the OV of two regional railway links in the Netherlands, in which an Internet-based SP survey was conducted, and logit models were established for estimating CS and OP [29]. In their study, the authors showed the classification of economic values (see Figure 1), although it should be noted that various classifications can be found in literature. Laird et al. (2009) reviewed past empirical studies on transport-related OVs and non-use values to demonstrate their importance in railway scheme appraisals [30]. Chang (2010) estimated OV and the non-use value of high-speed railway service and conventional railway service. A double-bound dichotomous choice survey was applied for collecting the data, and survival analysis was applied for estimating OV and non-use values [31]. Chang et al. (2012) estimated OV and the non-use value of bus services in South Korea [32]. They applied a utility function-based method using single-bound data and a survival analysis-based method incorporating double-bound data and compared them. Lee and Burris (2018) estimated OV of a high-occupancy vehicle lane in a freeway in South Korea based on the logsum concept [33].

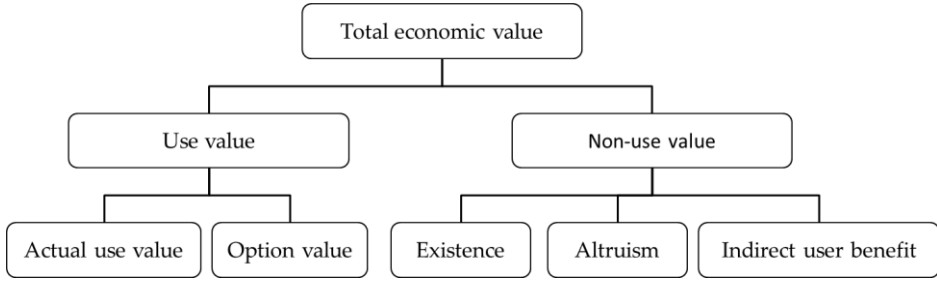

**Figure 1.** Classification of economic benefit categories [25].

As shown above, there are many studies that investigate OV of transportation services. However, almost all studies focus on services that have significant demand. On the other hand, local bus services in Japan are unprofitable; they account for a small share of the transport market. For example, in Aichi Prefecture, which is the central region of Japan, the aggregative bus share is only 0.5% [34]. Therefore, it is important to investigate the value of such unprofitable local bus services. In addition, this study compares the values of local bus services among different cities, while the studies above focused on one or two target services.

In literature, OV, the difference between OP and expected CS, is frequently investigated for evaluating the target service. OP value can be estimated as the willingness to pay for maintaining a target service by the contingent valuation method, which can be applied relatively easily. Its value can be estimated from the utility function of the discrete choice model [28,29,32] or from the value function using the probability density function approach [31,32,35]. The CS value is estimated using discrete choice analysis. That is, in literature, CS is measured from the non-use utility [28,29] or logsum value [33,36]. However, this methodology cannot be applied when the use frequency of a target transportation service is very low, since respondents do not consider such services a choice option. Thus, a reasonable parameter for the utility function cannot be obtained. Therefore, in this study, only OP for a local bus service was investigated.

## 3. Methodology

Smith (1985) argued that the uncertainty of maintaining a target service should be considered when evaluating OP [20]. Similarly, Freeman III (1986) considered the probability of a target service supply, as well as the probability of using a target service. In this study, the authors applied the concept of utility formulation in Freeman III (1986). The individual $n$'s utility of paying an additional monthly tax, $tax_n$, for maintaining a local bus service in his/her city, $U_{pay,n}$, and that of not paying it, $U_{not-pay,n}$, are formulated as follows:

$$U_{pay,n} = p_n q'_n u_n(tax_n, 1) + p_n (1 - q'_n) u_n(tax_n, 0) + (1 - p_n) \widetilde{u}_n(tax_n) \tag{1}$$

$$U_{not-pay,n} = p_n q_n u_n(0, 1) + p_n (1 - q_n) u_n(0, 0) + (1 - p_n) \widetilde{u}_n(0), \tag{2}$$

where $tax_n$ is the amount of tax that can equal zero and $p_n$ is the probability of individual $n$ being inclined to use the local bus service in a target time period (a month). $q_n$ is the probability that the local bus service is maintained in that period without subsidy. $q'_n$ is this probability with subsidy, and its value is larger than the probability without subsidy ($q'_n > q_n$). $u_n(tax_n, 1)$ is the sub-utility wherein the local bus service can be used when individual $n$ wants to use it, $u_n(tax_n, 0)$ is the sub-utility wherein it cannot be used when he/she wants to use it, and $\widetilde{u}_n(tax_n)$ is the sub-utility relating to the non-use value. These sub-utilities are formulated as follows:

$$u_n(tax_n, 1) = \alpha tax_n + \boldsymbol{\beta} \mathbf{x}_n + \boldsymbol{\beta}' \mathbf{x}'_n + \varepsilon_{1,n} \tag{3}$$

$$u_n(tax_n, 0) = \alpha tax_n + \boldsymbol{\beta} \mathbf{x}_n + \boldsymbol{\beta}' \mathbf{x}'_n + \gamma + \varepsilon_{0,n} \tag{4}$$

$$\widetilde{u}_n(tax_n) = \alpha tax_n + \boldsymbol{\beta}\mathbf{x}_n + \widetilde{\boldsymbol{\beta}}\widetilde{\mathbf{x}}_n + \widetilde{\varepsilon}_n, \tag{5}$$

where $\mathbf{x}_n$, $\mathbf{x}'_n$, and $\widetilde{\mathbf{x}}_n$ are vectors of their explanatory variables, $\alpha$, $\boldsymbol{\beta}$, $\boldsymbol{\beta}'$, and $\widetilde{\boldsymbol{\beta}}$ are vectors for unknown parameters, and $\varepsilon_{1,n}$, $\varepsilon_{0,n}$, and $\widetilde{\varepsilon}_n$ are error terms. $\alpha tax_n$ is the direct disutility for paying an additional tax and $\boldsymbol{\beta}\mathbf{x}_n$ is the other utility relating to tax payment. $\boldsymbol{\beta}'\mathbf{x}'_n$ is the utility for when individual $n$ is inclined toward using the local bus service and $\widetilde{\boldsymbol{\beta}}\widetilde{\mathbf{x}}_n$ is the utility that does not relate to this situation. $\gamma$ is the disutility for when the local bus service cannot be used when individual $n$ is inclined to use it. This disutility expresses the regret of individual $n$ and one possible expression is:

$$\gamma = -\exp(\boldsymbol{\mu}\mathbf{z}_n), \tag{6}$$

where $\mathbf{z}_n$ is the vector of the explanatory variables and $\boldsymbol{\mu}$ is the parameter vector.

Substituting Equations (3)–(5) into Equations (1) and (2), the total utility of paying an additional tax and that of not paying are expressed as follows:

$$U_{pay,n} = V_{pay,n} + \varepsilon_{pay,n} = \alpha tax_n + \boldsymbol{\beta}\mathbf{x}_n + p_n\boldsymbol{\beta}'\mathbf{x}'_n + (1 - p_n)\widetilde{\boldsymbol{\beta}}\widetilde{\mathbf{x}}_n + p_n(1 - q'_n)\gamma + \varepsilon_{pay,n} \tag{7}$$

$$U_{not-pay,n} = V_{not-pay,n} + \varepsilon_{not-pay,n} = p_n\boldsymbol{\beta}'\mathbf{x}'_n + (1 - p_n)\widetilde{\boldsymbol{\beta}}\widetilde{\mathbf{x}}_n + p_n(1 - q_n)\gamma + \varepsilon_{not-pay,n}, \tag{8}$$

where $V_{pay,n}$ and $V_{not-pay,n}$ are the systematic terms of each utility function and $\varepsilon_{pay,n}$ and $\varepsilon_{not-pay,n}$ are composite error terms, $\varepsilon_{pay,n} = p_n(1 - q'_n)\varepsilon_{0,n} + p_nq'_n\varepsilon_{1,n} + (1 - p_n)\widetilde{\varepsilon}_n$ and $\varepsilon_{not-pay,n} = p_n(1 - q_n)\varepsilon_{0,n} + p_nq_n\varepsilon_{1,n} + (1 - p_n)\widetilde{\varepsilon}_n$. Assuming that these composite error terms follow an independent normal distribution with a zero mean, the probability of paying a tax can be expressed by the following binary probit model:

$$\begin{aligned}P_n(pay) &= P_n(U_{pay,n} > U_{not-pay,n}) = Prob(\overline{\varepsilon}_n < V_{pay,n} - V_{not-pay,n}) \\ &= \Phi_1\left(\frac{V_{pay,n} - V_{not-pay,n}}{\sigma}\right) = \Phi_1\left(\frac{\alpha tax_n + \boldsymbol{\beta}\mathbf{x}_n + p_n(q_n - q'_n)\gamma}{\sigma}\right),\end{aligned} \tag{9}$$

where $\Phi_1(\cdot)$ is the cumulative density function of a univariate normal distribution and $\overline{\varepsilon}_n = \varepsilon_{not-pay,n} - \varepsilon_{pay,n}$ is the error term that follows normal distribution $N(0, \sigma^2)$.

To estimate unknown parameters, it is necessary to collect answers to question on the intention to pay tax for maintaining the local bus service. In the question, the tax level, as well as the probability that the local bus service is maintained with and without subsidy, must be shown to respondents. These values are explained in Section 4. The probability that individual $n$ will be inclined to use the local bus service in a month, $p_n$, is calculated from the bus use frequency of individual $n$. In this study, the authors focused on three typical trip purposes: Commuting, daily shopping, and doctor visits. When the bus use probability for each trip purpose is obtained, the bus use probability, $p_n$, is shown as follows:

$$p_n = 1 - \prod_m(1 - p_{m,n}), \tag{10}$$

where $p_{m,n}$ is the bus use probability for trip purpose $m$, $m$ = 1, 2 or 3. $p_{m,n}$ is calculated from the local bus use frequency by $p_{m,n} = \min(f_{m,n}, 1)$, where $f_{m,n}$ is the average bus use frequency for trip purpose $m$ in a month.

Alberini (1995) discussed the efficiency of a double-bound survey in the contingent valuation method [35]. In the double-bound survey, two answers are obtained from each respondent, and a bivariate binary probit model should be applied for the correlation between the two answers. For example, the probability that the individual $n$ answers "pay" for both questions is expressed as follows:

$$P_n(pay_1, pay_2) = \Phi_2\left(\frac{V_{1,pay,n} - V_{1,\ not-pay,n}}{\sigma}, \frac{V_{2,pay,n} - V_{2,not-pay,n}}{\sigma}, \rho\right), \tag{11}$$

where $\Phi_2(\cdot)$ is the cumulative density function of the bivariate normal distribution, $V_{i,pay,n}$ and $V_{i,not-pay,n}$ are systematic utility terms for the $i$th question, and $\rho$ is the correlation coefficient of the error terms. In the parameter estimation, the value of $\sigma$ is set to one without loss of generality.

From the above formulation, the OP, the maximum value of tax that individual $n$ can pay, is obtained, when $U_{pay,n}$ equals $U_{not-pay,n}$.

$$OP = \frac{-p_n(q_n - q'_n)\gamma - \boldsymbol{\beta}\mathbf{x}_n}{\alpha} \tag{12}$$

## 4. Data

This study applied a web questionnaire to collect data. The survey period was from December 2 to 5, 2016, and with a total sample of 940. The survey contained six groups of cities in five prefectures (Aichi, Mie, Shizuoka, Gifu, and Nagano) in central Japan, and the numbers of survey samples were controlled. In this survey, more samples were collected from smaller cities because the public bus companies there face more operational difficulties. Therefore, each city group's sample size was controlled by elaborately analyzing the data from small cities. The city group and its definition by population size are shown in Table 1. This classification is based on the Japanese city category approved by the prefectural governor and the Minister for Internal Affairs and Communications, Japan [37]. The Metropolis is Nagoya, which is the central city of the third largest metropolitan area in Japan.

**Table 1.** City groups and sample size.

| City Group | Definition: Population Size | Total Population [1] | Age Share [1] | | Sample Size |
|---|---|---|---|---|---|
| Metropolis | 2,260,440 | 2,295,638 | ≤19 years old | 17.2% | - |
| | | | 20–39 years old | 25.0% | 42 |
| | | | 40–59 years old | 27.9% | 42 |
| | | | ≥60 years old | 29.9% | 21 |
| | | | total | 100.0% | 105 |
| Major city | >500,000 | 1,502,969 | ≤19 years old | 17.6% | - |
| | | | 20–39 years old | 21.3% | 42 |
| | | | 40–59 years old | 27.0% | 42 |
| | | | ≥60 years old | 34.1% | 21 |
| | | | total | 100.0% | 105 |
| Core city | 200,000~500,000 | 3,732,676 | ≤19 years old | 18.9% | - |
| | | | 20–39 years old | 22.6% | 62 |
| | | | 40–59 years old | 27.2% | 62 |
| | | | ≥60 years old | 31.3% | 31 |
| | | | total | 100.0% | 155 |
| Medium city | 50,000~200,000 | 7,004,133 | ≤19 years old | 18.7% | - |
| | | | 20–39 years old | 21.9% | 62 |
| | | | 40–59 years old | 26.7% | 62 |
| | | | ≥60 years old | 32.7% | 31 |
| | | | total | 100.0% | 155 |
| Ordinary city | 20,000~50,000 | 1,748,441 | ≤19 years old | 18.2% | - |
| | | | 20–39 years old | 20.5% | 84 |
| | | | 40–59 years old | 26.2% | 84 |
| | | | ≥60 years old | 35.1% | 42 |
| | | | total | 100.0% | 210 |
| Small city | <20,000 | 846,148 | ≤19 years old | 16.4% | - |
| | | | 20–39 years old | 17.4% | 84 |
| | | | 40–59 years old | 25.0% | 84 |
| | | | ≥60 years old | 41.3% | 42 |
| | | | total | 100.0% | 210 |

[1] Total population and age share were obtained from the national population census in 2015.

The survey respondents should meet three conditions. First, respondents should live in a city with an operating bus line. Second, residents can call a taxi by phone from their residence. Third, the respondent is older than 20 years. The second condition is set for the investigation of the OP and OV of taxi services which are another important transport service for the elderly, although it will be reported at another opportunity. The reason for the third condition is that the respondents who are older than 20 years are obliged to pay taxes in Japan. In addition, the age of the respondents was also controlled. From the age of 20 onward, five age groups were set at every ten-year interval (the oldest group is over 60 years) and each group had the same sample share (20%). Although the elderly were defined as individuals who are 65 years or older, the oldest age group is unfortunately more than 60 years old because of the restriction of web-survey monitors. This age group will be treated as an elderly group in the following analysis.

Table 2 summarizes the data with a total sample size of 940 respondents. The largest portion among all occupations was company employees at 36.2%, and 93% of respondents had a vehicle license. Regarding household composition, almost one-third of all households included one or more elderly members. Respondents were also required to provide information about family vehicle ownership. A total of 91.9% of respondents answered that their family had one or more vehicles.

**Table 2.** Summary of survey respondents' basic statistics.

| Information | Description | Data size | Percentage |
| --- | --- | --- | --- |
| Gender | Male | 476 | 50.6% |
| | Female | 464 | 49.4% |
| Occupation | Employee | 340 | 36.2% |
| | Civil servant | 61 | 6.5% |
| | Self-employed | 77 | 8.5% |
| | Student | 18 | 1.9% |
| | Stay-at-home spouse | 182 | 19.4% |
| | Part-time job and freelancer | 136 | 14.5% |
| | Unemployed | 111 | 11.8% |
| | Others | 15 | 1.6% |
| License | Have a license | 874 | 93.0% |
| | Do not have a license | 66 | 7.0% |
| Household composition | Family with elderly person older than 65 years | 324 | 34.5% |
| | Family with young person, younger than 15 years | 304 | 32.3% |
| | Family without elderly or young people | 332 | 35.3% |
| Age group | Young age (20–39) | 376 | 40.0% |
| | Middle age (40–59) | 376 | 40.0% |
| | Elderly (60–) | 188 | 20.0% |
| Vehicle ownership | Household with one or more vehicles | 864 | 91.9% |
| | Household without a vehicle | 76 | 8.1% |

Table 3 shows the average values of bus fare paid for the current bus service. This information was also collected by the web questionnaire. From the table, it can be found that there is a clear tendency of bus fare among city groups. That is, residents in smaller cities tend to pay higher bus fares. This implies that residents in smaller cities have to travel longer distances than those in larger cities.

**Table 3.** Average bus fare.

| City Group | Commuting (JPY) | Shopping (JPY) | Doctor's Visit (JPY) |
| --- | --- | --- | --- |
| Metropolis | 157.1 | 183.6 | 156.9 |
| Major city | 296.7 | 175.2 | 191.1 |
| Core city | 307.4 | 204.2 | 230.0 |
| Medium city | 253.1 | 230.8 | 367.1 |
| Ordinary city | 319.6 | 262.7 | 324.3 |
| Small city | 341.0 | 322.0 | 398.7 |

The questionnaire consisted of five sections, as displayed in Table 4. The first section collected personal and family characteristics; these data are shown in Table 2. The second section collected service-level information for the local bus services. The third section collected respondents' actual travel behavior for three typical trip purposes: Commuting, shopping, and doctor visits. The fourth section collected bus use frequencies for the three typical trip purposes, $f_{m,n}$. The fifth section asked the intention of paying an additional tax for maintaining the local bus service, and the answers in this section were used for estimating OP.

**Table 4.** Survey contents.

| Section | Contents |
| --- | --- |
| I. Personal and family characteristics | Age, gender, occupation, driver's license, etc. Household composition, vehicle ownership, family income, etc. |
| II. Local public transport service level | Distance from the residence to the nearest bus stop, bus operating interval, distance from the residence to the nearest railway station, etc. |
| III. Daily travel behavior | Trip frequencies for three typical trip purposes (commuting, shopping, and doctor's visits), travel mode, travel time, etc. |
| IV. Bus use frequency | Local bus use frequencies for three typical trip purposes (commuting, shopping, and doctor's visits) |
| V. Tax paying intention | Double-bounded questions of tax paying intention for maintaining local bus service in respondent's city. |

In the fifth section, the respondents were given a hypothetical situation, and then asked about their intention to pay twice. This hypothetical situation is: "The local bus company in your city is facing financial difficulty and will stop the service with A% of probability. If all adult people in your city pay an additional tax, B JPY, every month, the local bus service will be maintained. This tax will be collected and managed by the local government. Currently, C% of citizens of your city agree to this additional taxation policy." In this scenario description, two probability variables in the utility functions—$q_n$ and $q'_n$, which are the probabilities that the local bus service is maintained without and with subsidy—are 1.0-A/100 and 1.0, respectively. The setting of values for A, B, and C, and the double-bounded asking flow, are shown in Figure 2.

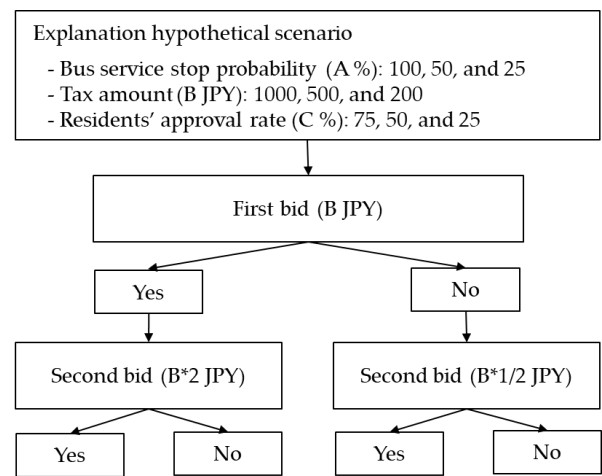

**Figure 2.** Flow of double-bounded question for tax paying intention survey.

## 5. Estimation Result and Discussions

Table 5 displays the estimation results of the tax paying model. Since it is not possible to estimate the model parameters for all combinations of age and city groups because of the limited sample size, three models were estimated for three age groups; the difference in three city groups was considered by introducing different constant terms. The explanatory variables of the bus service level, service frequency, and distance to nearest bus stop were retained if they had a reasonable sign of the parameter value and even if their parameters were not significant statistically. Regarding other parameters, except for constants, the only parameters that were almost significant statistically were retained. As a result, social interaction—residents' approval rate (C%)—and variables regarding service level of rail were not remained.

**Table 5.** Estimation results of the tax paying intention model.

| Explanatory Variables | | Age 20–39 | | Age 40–59 | | Age 60– | |
|---|---|---|---|---|---|---|---|
| | | Estimates | t-Stat. | Estimates | t-Stat. | Estimates | t-Stats |
| $tax_n$ (1000 JPY) | | −0.507 | −3.57 *** | −0.478 | −4.47 *** | −0.288 | −1.79 * |
| $\beta x_n$ | | | | | | | |
| Constant (Metropolis and Major city) | | −0.383 | −2.09 ** | −0.397 | −1.77 * | 0.0205 | 0.07 |
| Constant (Core and Medium city) | | −0.216 | −1.45 | −0.140 | −0.80 | 0.215 | 0.95 |
| Constant (Ordinary and Small city) | | −0.149 | −1.10 | −0.233 | −1.59 | 0.215 | 1.09 |
| Male | | | | 0.231 | 1.90 * | | |
| Service frequency at nearest bus stop (number of services/hour in daytime) | | | | 0.0577 | 1.80 * | | |
| Distance to nearest bus stop (km) | | −0.0295 | −1.37 | | | −0.0105 | −0.28 |
| Occupation | Civil servant | | | | | 0.879 | 1.92 * |
| | Self−employed | | | −0.586 | −2.75 *** | | |
| | Part−time | −0.253 | −1.72 * | | | | |
| $\gamma$ | | | | | | | |
| Constant (Metropolis and Major city) | | −0.403 | −0.91 | 0.626 | 1.09 | 0.213 | 0.54 |
| Constant (Core and Medium city) | | −0.975 | −1.76 * | 0.536 | 0.87 | −3.48 | −0.30 |
| Constant (Ordinary and Small city) | | −0.854 | 1.90 * | 0.814 | 1.42 | −0.844 | −0.80 |
| Company worker | | 0.515 | 1.78 * | | | | |
| Driving license | | | | −0.971 | −1.84 * | | |
| Family annual income (million JPY) | | 0.0759 | 1.61 | | | | |
| Correlation ($\rho$) | | 0.363 | 3.32 *** | 0.840 | 15.19 *** | 0.652 | 6.06 *** |
| Sample size | | 376 | | 376 | | 188 | |
| Log-likelihood LL(0) | | −521.2 | | −521.2 | | −260.6 | |
| Log-likelihood LL($\beta$) | | −455.1 | | −445.7 | | −237.4 | |
| Adjusted $\rho^2$ | | 0.104 | | 0.120 | | 0.047 | |

\* $p < 0.1$, \*\* $p < 0.05$, \*\*\* $p < 0.01$.

The response to the amount of tax is the most important in this study. As Equation (12) shows, the parameter value for the tax directly influences on the value of OP. The parameters for tax are significant in all age classes. Hence, the amount of tax is important information in tax payment decisions. The younger residents tended to show the strongest resistance to a larger amount of tax, while elderly residents showed the smallest. Thus, younger residents were not concerned about the social issue of the local bus.

The other parameter estimates show that the service level of the bus influenced decision-making only for the middle-age residents. Young part-timers and middle-age self-employed residents tended to oppose tax payments for maintaining the local bus service. In the regret term, $\gamma$, young company workers tended to feel regret, while middle-age individuals with driving licenses did not, when they cannot use the local bus service.

To ensure the validity of the applied model, the authors compared the final log-likelihood values with the bivariate binary probit model with usual linear utility function, in which all explanatory variables shown in Table 5 were incorporated. The log-likelihood values by linear utility function were $-470.5$, $-454.3$, and $-240.0$ for ages 20–39, 40–59, and $\geq$60, respectively. Since these numbers are lower than those by the applied model, the applied model can be validated.

The fitness indexes show that high fitness could not be obtained, especially in the elderly group. In fact, only a limited number of parameters were introduced into the models. One reasonable explanation is that many respondents in each age group showed similar responses; hence, significant diversity in each age group did not exist. For example, many respondents in the elderly group had no occupation.

Table 6 shows the estimated OP values. In the calculation of these values, actual population distribution in each city group was considered. This table shows that Metropolis has the highest OP value and Major city has the second highest OP value. There is no clear tendency in OP value from Core city to Small city. Compared with the bus fares which were shown in Table 3, OP values were much higher than bus fares. The difference between current bus fare and OP was bigger in larger cities. To evaluate the appropriate amount of subsidy, it is necessary to calculate CS and OV. However, the local bus service in small cities offers a low level of service, so the CS was expected to be low. For example, Bristow et al. (1992) showed that CS tends to have around half the value of OP in an unprofitable local bus in the U.K. In addition, Masuoka (2004) presented a case wherein the deficit of a local bus company in a small city with 45,000 residents was around 20 million JPY [38]. The results of these studies show that a small local bus company in a small city with only 10,000 adults should be subsidized with 2.8 million JPY (half of OP) every month, and this subsidy can cover its deficit.

**Table 6.** Estimated option price.

| City Group. | Option Price (JPY/month) | Age Group | Option Price (JPY/month) |
|---|---|---|---|
| Metropolis | 995.2 | | |
| Major city | 743.2 | | |
| Core city | 569.5 | 20–39 years old | 218.2 |
| Medium city | 490.2 | 40–59 years old | 474.2 |
| Ordinary city | 638.7 | $\geq$60 years old | 1001.9 |
| Small city | 569.4 | | |

In terms of OP value for each age group, a clear tendency was evident. Younger residents showed the smallest OP value and elderly residents showed the highest. In fact, the OP value of elderly residents was more than double that of middle-age residents and more than fourfold over that of younger ones. This result indicates that the elderly were seriously concerned about the local bus problem. Thus, the local bus service should be maintained in an aging society.

## 6. Conclusions

This study focused on the OP of local bus services and compared its value among different city-size groups and age groups. Double-bounded question data were collected, and the tax payment model based on the concept proposed by Freeman III (1986) was established for estimating OP value. The result showed that the older age group showed least resistance to paying taxes to maintain local bus services. This study, to the best of the authors' knowledge, is the first to compare OVs among city and age groups and to apply the Freeman III concept.

The results showed that the amount of tax is important to the tax payment decision for maintaining the local bus service. Citizens in larger cities tended to show higher OP values. In addition, the OP value of the elderly was more than double that of the middle-age residents and more than fourfold over that for younger ones. This result indicates that the elderly are seriously concerned about the local bus problem. Thus, the local bus service should be maintained in an aging society and the local government should consider the subsidization to local bus service, although in Japan, the OV of transport service is not explicitly considered in cost–benefit analyses in transportation planning. That is, more substantial subsidization to local bus companies can be justified by the examination of the OV and OP of such services.

Future research should attempt to estimate consumer surplus of local bus services and evaluate the option value accordingly. In addition, other transportation modes for the elderly, such as taxi, should be investigated for a prospective super-aging society.

**Author Contributions:** We thank X.Z. and T.M. for helping with the conceptualization of the study, X.Z. and T.M. for developing the methodology, Gauss 4.0. developers for their software, and T.M. for the writing—review and editing.

**Funding:** This research was financially supported by the Japan Society for the Promotion of Science through the Grant-in-Aid for Scientific Research (no. 16K12825), and the authors gratefully acknowledge this support.

**Conflicts of Interest:** The authors declare no conflict of interest.

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
