# Peer review of "A Comparative Analysis on Residents’ Reservation Willingness for Bus Service Based on Option Price"

_sustainability, doi:10.3390/su11010260_

Round 1
Reviewer 1 Report
p.p1 {margin: 0.0px 0.0px 0.0px 0.0px; font: 11.0px 'Helvetica Neue'; color: #000000; -webkit-text-stroke: #000000} p.p2 {margin: 0.0px 0.0px 0.0px 0.0px; font: 11.0px 'Helvetica Neue'; color: #000000; -webkit-text-stroke: #000000; min-height: 12.0px} span.s1 {font-kerning: none}Report on ‘A comparative analysis on residents’ reservation willingness for bus service based on option price’ by Xun Zheng and Tomio Miwa
The authors imply the options price that various demographical groups are willing to pay for maintaining local bus services. They use the contingent valuation method in conjunction with a double-bound survey in which they collected data from people in different age groups (young to old) living in different types of cites (small to large). From this they obtain the option value that each demographic group would be willing to pay to maintain the bus service. Elderly people are willing to pay more.
The paper is interesting and I like the methodology in principle. However it is difficult to judge if the complexity of the model is warranted, or if similar results could be obtained in a much simpler way. The paper would benefit around some such discussion, for example of you just take the mean of how much each demographic says they are willing to pay do you get similar results? What is the added value by using the current approach?
I think a discussion and some study around this would be necessary in order to validate the merit of the paper.
Author Response
Firstly, we would like to thank the area editor and the reviewers for their precious time and invaluable comments. We have carefully addressed all the comments. The corresponding changes and refinements made in the revised paper are summarized in the attached file. If there is any further questions or comments, please don't hesitate to tell us.

Reviewer 2 Report
Dear Authors,
This is an interesting study with practical implications relevant for many societies that face similar problems with ageing population. Please take into account the following suggestions.
There is no reference in your paper to sustainability. This focuses simultaneously on three fields: economic, social, and environmental. The last one is missing in your paper. Please explain how your study can improve sustainability.
L 169 “Second, residents can call a taxi by phone from their residence.” Why do you have this condition?
L 170 “Third, the respondent is older than 20 years.” Why 20 and not 18 or 22?
L 169-170. “Second, residents can call a taxi by phone from their residence. Third, the respondent is older than 20 years. The second condition is…” Which is, in fact, the second one?
L 179-180. “Regarding household composition, almost one-third of all households include one or more elderly and/or younger members.” As it is formulated now and without other explanations, the sentence covers all possibilities. Please correct it.
What is the current price paid for public transportation?
Current tariffs also influence WTP, besides the factors that you have taken into account (bus service stop probability, residents’ approval rate). How did you include this influence in your analysis?
Have you taken into account the influence of other variables on WTP, such as access to alternative solution? If yes, how? If no, why?
What are the practical implications of your findings? What solutions can be suggested?
Author Response
We would like to thank the area editor and the reviewers for their precious time and invaluable comments. We have carefully addressed all the comments. The corresponding changes and refinements made in the revised paper are summarized in the attached file. If there is any further questions or comments, please don't hesitate to tell us.

Round 2
Reviewer 1 Report
The authors new version is better, and I recommend publication.
Author Response
Thank you for reviewing.
Reviewer 2 Report
Dear Authors,
The manuscript increased in clarity, especially in some parts related to characteristics of studied area, which are usually unknown to readers from other regions.
However, there are several main aspects mentioned last time that still need to be addressed properly:
1. “There is no reference in your paper to sustainability. This focuses simultaneously on three fields: economic, social, and environmental. The last one is missing in your paper. Please explain how your study can improve sustainability.”
The introduction should include at least references to the following: Why is your study relevant to sustainability research? What is its relevance in the context of similar scientific literature?
Results and discussion should highlight its contribution to sustainability.
Conclusions should also make reference to sustainability.
Please consider that you submitted your paper to Sustainability Journal.
2. “What are the practical implications of your findings? What solutions can be suggested?”
The given answer is rather shallow. Please try to explain who can use your results and how in a more detailed manner because this is the final purpose of most research in this field – the practical application of its results.
Author Response
Thank you for the second round of Reviewers’ comments and the opportunity to further revise the paper. It is a high priority for us to arrange the manuscript into a most understandable form. We appreciate the time and effort the reviewers and editors have invested in assisting with our paper. The specific changes made in response to the reviewer comments are detailed in the response report.
